# Reproducibility study for "Explaining in Style: Training a GAN to explain a classifier in StyleSpace"

## Reproducibility Summary

**Scope of Reproducibility**

This work aims to reproduce Lang et al.'s StylEx [9] which proposes a novel approach to explain how a classifier makes its decision. They claim that StylEx creates a post-hoc counterfactual explanation whose principal attributes correspond to properties that are intuitive to humans. The paper boasts a large range of real-world practicality. However, StylEx proves difficult to reproduce due to its time complexity and holes in the information provided. This paper tries to fill in these holes by: i) re-implementation of StylEx in a different framework, ii) creating a low resource training benchmark.

**Methodology**

We use their provided python notebook to confirm their *AttFind* algorithm. However, to test the authors' claims, we reverse engineer their architecture and completely re-implement their train algorithm. Due to the computational cost of training, we use their pre-trained weights to test our reconstruction. To expedite training, a smaller resolution dataset is used. The training took 9 hours for 50,000 iterations on a Google Colab Nvidia K80 GPU. The hyperparameters are listed in the proceedings.

**Results**

We reproduce the StylEx model in a different framework and test the *AttFind* algorithm, verifying the original paper's results for the perceived age classifier. However, we could not reproduce the results for the other classifiers used, due to time limitations in training and the absence of their pre-trained models. In addition, we verify the paper's claim of providing human-interpretable explanations, by reproducing the two user studies outlined in the original paper.

**What was easy**

The notebook supplied by the authors loads their pre-trained models and reproduces part of the results in the paper. Furthermore, their algorithm for discovering classifier-related attributes, *AttFind*, is well outlined in their paper making the notebook easy to follow. Lastly, the authors were responsive to our inquiries.

**What was difficult**

A major difficulty was that the authors provide only a single pre-trained model, which makes most of the main claims require training code to verify. Moreover, the paper leaves out information about their design choices and experimental setup. In addition, the authors do not provide an implementation of the models' architecture or training. Finally, the practical audience is limited by the resource requirements.

**Communication with original authors**

We had modest communication with the original author, Oran Lang. Our discussion was limited to inquiries about design choices not mentioned in the paper. They were able to clarify the encoder architecture and some of their experimental setup. However, their training code could not be made available due to internal dependencies.

# 1 Introduction

As the field of machine learning (ML) develops and its algorithms become more prevalent in society, concerns on the explainability of black-box models become pivotal. For problems that have a high societal impact, there is understandable apprehension towards trusting models that do not provide justification. For applications such as medical imaging and autonomous driving, there is a need for some level of human supervision. Even if a model has high performance, such as neural networks, without the ability for human interpretation, its use will be limited.

In order to gain trust in systems powered by ML models, the models need to be interpretable and explainable. The two concepts are regularly used interchangeably, yet have subtle differences. Interpretability is the degree to which humans can understand the cause of a decision [10]. Deep neural networks, such as classifiers are often perceived as "black boxes" whose decisions are opaque and hard for humans to understand. Explaining the decision of classifiers can reveal model biases[8] and also provide support to downstream human decision-makers. On the other hand, explainability is linked to the internal logic of a model. It focuses on explaining the data representation within that network. Explainability implies interpretability, however, the implication is not bidirectional.

In recent years, there has been increasing attention to the field of explainability of deep network classifiers. Among the various ways of explanations, counterfactual explanations are gaining increasing attention [11, 2, 3]. To discover and visualize, the attributes used to generate counterfactual explanations, a natural candidate is generative models. In [13] they observed that StyleGAN2 [7], tends to contain a disentangled latent space (i.e., the "StyleSpace") which can be used to extract individual attributes. The authors based their proposed methodology [9] on this observation. Though [12] propose a similar architecture, Lang et al. assert that by integrating the classifier into the training of StylEx they can obtain principal attributes that are specific for the classification task. Additionally, they suggest that StylEx can be applied to a large variety of complex, real-world tasks, which makes its replicability especially intriguing.

Our work aims to reproduce the claims made by Lang et al. and confirm their results. Their paper reports in detail many experiments to justify their claims, but does not dive into their experimental setups for architecture and training. Since not all the information needed is available without contacting the authors, we argue that this paper cannot be considered *fully* reproducible.

To remedy the holes in reproducibility and aid future work that builds on or applies StylEx, we build their proposed architecture and training algorithm, after correspondence with the authors.

# 2 Scope of reproducibility

To determine the scope of reproduction, we quote Lang et al.'s main claims:

Claim 1 : [They] propose the StylEx model for classifier-based training of a StyleGAN2, thus driving its StyleSpace to capture classifier-specific attributes

Claim 2 : A method to discover classifier-related attributes in StyleSpace coordinates, and use these for counterfactual explanations.

Claim 3 : StylEx is applicable for explaining a large variety of classifiers and real-world complex domains. [They] show it provides explanations understood by human users.

To reproduce Claim 2, a trained model and the *AttFind* algorithm are sufficient; both of which are contained in the authors' notebook. Claim 1 requires a network trained conditioned on a classifier and a network trained without, while Claim 3 requires multiple networks trained on multiple domains. However, to train these models, the architecture and training code is necessary; which, as stated previously, are not open source or thoroughly documented. In addition, the computational cost to train the models is expensive. Thus, to verify these claims our goals will be to:

- Reconstruct their architecture and port the pre-trained weights in PyTorch

- Evaluate whether the principal attributes we obtain correspond to the same features using their pre-trained weights

- Retrain on datasets of smaller images and analyze the scalability of their method using fewer training steps and smaller architecture

- Conduct two user studies on visual coherence and distinctness to prove that attributes extracted are interpretable by humans

To ease reproduction for future work, we built the StylEx architecture into a different framework, to get a deeper understanding of the model, and become more equipped to tackle training. As an addition, this contribution allows StylEx to be more accessible for classifiers trained in PyTorch.

## 3 Background

There have been many attempts to extract explanations from classifiers most of which utilize heatmaps of important features. However, heatmaps struggle to visualize features that are not spatially localized such as color or shape. Rather than identifying areas of interest, one can provide an explanation through a "what-if" example where the features are slightly altered. These forms of justification have been found to be more interpretable for non-localized features, and are known as *counterfactual* examples. However, it often requires domain knowledge and handcrafting examples to be appropriate. Lang et al. automate this and utilize machine learning to generate realistic counterfactual examples. This section will outline how they claim to achieve this with their two major contributions, StylEx and AttFind.

### 3.1 StylEx

The way Lang et al. generate examples is through a neural generative model they dubbed StylEx. StylEx expands on the popular generative adversarial network StyleGAN v2, which generates realistic images by creating competition between two networks.

One of these two networks, referred to as the Generator, $G$, attempts to generate a realistic image. To this end, the generator samples from a latent space, $z \in R^n$, with a simple probability distribution such as $z_i \sim \mathcal{N}(0, 1)$. The sampled vector is pushed through a series of linear layers called *mapping network* to create a new latent vector, $w$, with a more complex probability distribution. This vector is used as input to a number of *StyleBlocks* based on the logarithmic resolution of the image. StyleBlocks consist of an affine transform and an upsampling layer. The affine transform, $A_r$, maps $w$ to yet another vector $s_r$, where $r$ denotes the block number or resolution of the block. This concatenation of all $s_r$ is known as the style, or *attribute*, vector, and the space that it spans is known as the StyleSpace. The attribute space is emphasized due to recent observations that it is less entangled than the latent space. The second network is the discriminator, $D$. This network is trained to differentiate between fake and real images. This forces the generator to slowly improve its creation of fake images. In this way, the discriminator can be seen as an adaptive loss function.

The flaw with the direct application of StyleGAN is that it generates from a random latent space. To explain a classification, we would like to condition it on a particular image of interest, but StyleGAN has no mechanism for extracting the attributes of an image. To fix this, Lang et al. added a third, encoding network to StylEx, $E$. Rather than using a randomly sampled $z$ and the mapping network to obtain $w$, StylEx uses the output of the encoder, $z = E(x)$, where $x$ is an input image. StylEx adds an extra loss condition that the reconstructed image, $x' = G(E(x))$, should be approximately $x$. Thus, the encoder combined with the affine transformations allows us to extract the attributes of an input image.

StylEx is not unique in adding an encoder to the StyleGAN to explain a classifier. However, other methods do not include the classifier in the training of the network. StyleGAN incorporates the classifier into training by appending its output to the encoded $z$ vector. This results in another loss condition $C(x) \approx C(x')$.

### 3.2 AttFind

Once the attributes of an image have been extracted, a counterfactual explanation can be achieved from the attributes with the most affect on a classifier's decision. Lang et al. propose attribute find (AttFind) to discover the most influential attributes. The algorithm adjusts all the attributes one at a time by a fixed amount $d$ and observes their effect on the classification $\Delta c_s$. The $k$ attributes with the highest $\Delta c$ create a *local* explanation for an image's classification. To approximate a *global* explanation, the principal attributes are determined by the mean $\Delta c$ across images in a set.

## 4 Reproduction approach

Reimplementing StylEx has been split into two main tasks to ease resource requirements. The first task consists of rebuilding StylEx in a different framework; the second is training the model from scratch. In this section, we discuss how we rebuilt the model architecture and training process. Additionally, we include details obtained through correspondence missing from the original paper.

### 4.1 Model descriptions

To test Claim 1 and Claim 3, at least two models are necessary. Because only one pre-trained model is available, a new model needs to be trained. However, this is computationally expensive as it builds on StyleGAN [1]. This led us to evaluate reproducibility in two ways. Firstly, we recreate their architecture in PyTorch, using their pre-trained weights to bypass the training limitation. Secondly, we attempt to train a model from scratch using less complex datasets with smaller resolutions to verify claims requiring multiple models. In the following sections, we explain how we reconstruct the StylEx architecture and training process.

#### 4.1.1 Rebuilding StylEx

The author's notebook includes a TensorFlow StylEx pre-trained on the FFHQ[6] dataset to find the attributes most influential in age classification.

Taking advantage of the pre-trained model's raw parameters, we reverse engineer the architecture of each component of StylEx and implement it in PyTorch. Subsequently, the pre-trained weights are transferred into the reconstructed StylEx to confirm the correct implementation of the structure. Transferring the pre-trained parameters from a TensorFlow model to a PyTorch model turned out to be challenging and non-trivial.

We start by building the architecture of the MobileNetV1 [5] classifier, as described in the summary of their model, in both TensorFlow and PyTorch. We follow this approach so that we can compare how the results of each layer differ, depending on the framework. We notice that for the 2D convolutional layers PyTorch and TensorFlow pad the images differently, leading to different results. To address this, we add a ConstantPad2D layer in our PyTorch architecture before each convolution with a stride of 2. In addition, we change the default hyperparameters of PyTorch's BatchNorm2D to match the corresponding TensorFlow defaults.

The next step is to follow the same procedure for the encoder and the StyleGAN components. We use the official StyleGAN2 implementation in PyTorch by NVlabs[7] and modify the initial architecture to align with the StylEx model. In particular, instead of only using the encoding of an image $X$ as input to the generator, we also concatenate the classifier's output logits. Additionally, their generator returns the StyleSpace which contains classifier-specific attributes. For the encoder, we use the same architecture as StyleGAN2's discriminator. Finally, we transfer the pre-trained weights, to our components.

The last step is to load the rebuilt StylEx model in the provided notebook to confirm that the conversion of the models is successful and reproduce the results provided in the notebook.

#### 4.1.2 Training the model

Lang et al. asserted that StylEx works for a wide range of classifiers and datasets. The results they show in their paper are all with high-resolution images. The high resolution comes with a high computational cost as StylEx is built on top of a StyleGAN. High-resolution StyleGANs can take over a month to train on a single GPU system. To tackle this, we train our model on a low-resolution MNIST dataset. In this way, we investigate whether their model works well on low-resolution datasets and relieve computational requirements.

The training is as outlined in their paper. The loss function for the StylEx model is broken into seven parts: $\mathcal{L}_x$, $\mathcal{L}_w$, $\mathcal{L}_{LPIPS}$, $\mathcal{L}_{adv}$, $\mathcal{L}_{PLR}$, $\mathcal{L}_{KL}$, and the $\mathcal{L}_{GP}$. $\mathcal{L}_x$ is the L1 loss between the real image, $x$, and the reconstruction of that image, $G(E(x))$. $\mathcal{L}_{LPIPS}$ is the Learned Perceptual Image Patch Similarity (LPIPS) of the two images. This loss is a metric other than raw pixel value error for the similarity between two images. $\mathcal{L}_w$ is the L1 loss between the encoding of the original image, $w = E(x)$, and the encoding of the reconstructed image $w' = E(G(E(x)))$. Collectively, these three losses make up the reconstruction loss, $\mathcal{L}_{rec}$, ie,

$$\mathcal{L}_{rec} = \mathcal{L}_w + \mathcal{L}_x + \mathcal{L}_{LPIPS}.$$

In the implementation, each loss term in $L_{rec}$ had a weighting coefficient to even out the magnitude of their contributions. The weights are detailed further in Section 5.2.

$\mathcal{L}_{KL}$ is the KL divergence loss between the classification probabilities of the original image and its reconstructed classification probabilities. $\mathcal{L}_{GP}$ and $\mathcal{L}_{PLR}$ are the *gradient penalty* and *path length regularization* losses described in the WGAN-GP[4] and StyleGAN2 paper[7] respectively. $\mathcal{L}_{adv}$ is the Wasserstein adversarial generator loss of $x'$. Finally, the discriminator's loss is the Wasserstein adversarial discriminator loss.

---

[1]StyleGAN can take on the order of 40 days on one GPU for high resolutions [6]

## 5 Experimental setup

### 5.1 Datasets

The pre-trained models the authors offer are trained on the Flickr-Faces-HQ Dataset [6] [2]. The dataset contains 70,000 high-quality PNG images at 1024×1024 resolution with large variations in terms of age, ethnicity, and image background. They use it to find the top attributes which contribute to perceiving a person's age (young or old) or gender (male or female). They also preprocess the images by lowering the resolution to 256x256. The official dataset is unlabeled. It is not clear whether the authors' dataset is an internal, labeled Google version or an unofficially labeled dataset.

For training, the MNIST [1] dataset is used due to its simplicity. Only the examples with labels 8 or 9 are kept and the resolution is increased to 32x32. MNIST was chosen because images compressed to 16x16 or even 8x8 tend to be recognizable for humans. Unfortunately, LPIPS relies on neural networks that have a fixed number of pooling layers. Without editing reimplementation of LPIPS, the lowest resolution possible is 32.

### 5.2 Hyperparameters

A complete list of hyperparameters can be found in Table 2 (see Appendix C). A hyperparameter search was not performed for two reasons. First, the training time is long – even for very low resolutions, this is constraining. Second, the criteria for evaluating success is based on a human user, making automated hyperparameter tuning unintuitive.

### 5.3 Computational requirements

Most of our experiments were conducted on Google Colab along with our systems. For training our models we use Colab's NVIDIA Tesla K80 GPU. Our code is provided in the following GitHub repository: MLRC_2021_FALL-E358.

The basic architecture of the StyleGAN2 was adapted from NVlabs' GitHub repository. As previously mentioned, we modify the basic architecture, to align with StylEx's generator and load Lang et al.'s pre-trained weights. The training code was adapted from labml.ai Annotated Paper Implementations' StyleGAN implementation.

Training the model on MNIST for 50,000 iterations takes on the order of nine hours to train on Colab. The time required for AttFind is dependent on the resolution, latent dimension, and the number of images in the dataset. Finding the attribute of a single image took approximately one minute for an image with resolution 32 and a latent space of 514.

## 6 Results

### 6.1 Rebuilding StylEx results

To support Claim 1, we recreate their pre-trained models to PyTorch and test if our results agree. In Figure 3 (see Appendix A), we compare the results from our PyTorch StylEx to their TensorFlow implementation. There are minor differences in the probabilities from the PyTorch classifier which are likely caused by differences in default values or module implementations in the two frameworks.

### 6.2 AttFind results

We are now equipped to test our PyTorch models on the AttFind method and inspect the principal attributes of the age classifier; meaning the attributes with the highest contribution to young or old classification. To this end, we compute the AttFind algorithm – with our classifier and generator as inputs – using the 250 latent variables of the FFHQ dataset. As can be seen in Figures 1 and 5 (see Appendix B), our model obtains the same attributes as in the original paper.

In addition, we implement the **Independent** selection strategy, to generate image-specific explanations as described in the original paper. This method is a *local* explanation that returns the top-k attributes affecting a classifier's decision for a single image rather than the entire dataset. The results are shown in Figure 2.

These results support the author's Claim 2, that AttFind discovers significant attributes for a classifier's decision. Notably, in 1c the reported probability of the top left image is 17% in the paper, while the probability we find with our and their notebook classifier is 39%.

---

[2]https://github.com/NVlabs/ffhq-dataset

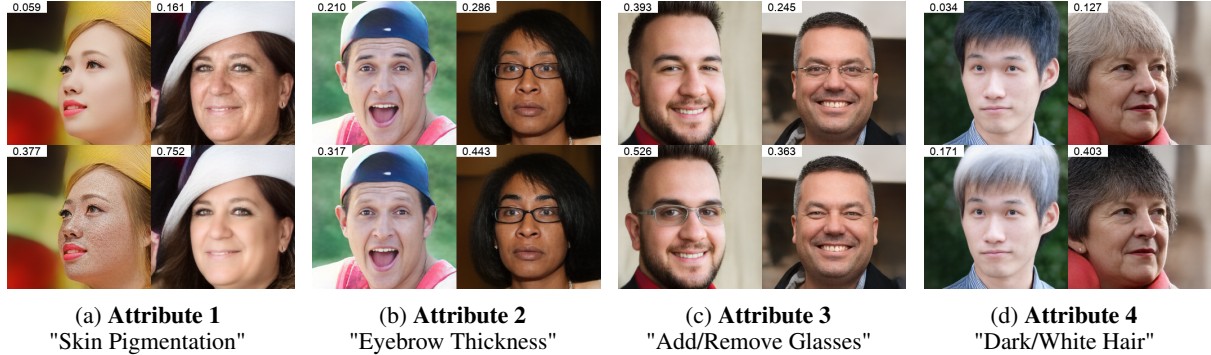

| (a) **Attribute 1** | (b) **Attribute 2** | (c) **Attribute 3** | (d) **Attribute 4** |
|---|---|---|---|
| "Skin Pigmentation" | "Eyebrow Thickness" | "Add/Remove Glasses" | "Dark/White Hair" |

Figure 1: **Top 4 attributes for the perceived age classifier detected by our model.** These images show how the probability of classifying a person as young or old changes based on each attribute. On the first column of each image we display the probability of the person being classified as old and on the second column the probability of them being classified as young.

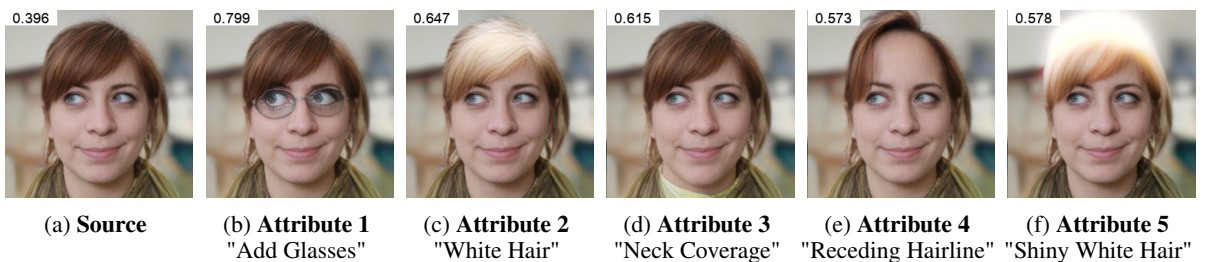

| (a) **Source** | (b) **Attribute 1** | (c) **Attribute 2** | (d) **Attribute 3** | (e) **Attribute 4** | (f) **Attribute 5** |
|---|---|---|---|---|---|
| | "Add Glasses" | "White Hair" | "Neck Coverage" | "Receding Hairline" | "Shiny White Hair" |

Figure 2: **Independent selection strategy.** Top-5 detected attributes for explaining a perceived-age classifier for a *specific* image. The attributes obtained are different from those presented in Figure 1 which are computed based on the largest average effect over 250 images. The probabilities displayed correspond to the person being classifier as old.

## 6.3 Quantitative evaluation results

To validate the authors' Claim 3 that attributes obtained are identifiable by humans, we conduct the two user studies explained in the paper. Both studies (Classification and Verbal description) aim to prove that the top extracted attributes are distinct, visually coherent, and can be used as counterfactual explanations.

The material used for the classification study was obtained by our PyTorch StylEx model on the perceived gender classifier (top 6 attributes), and by the authors' supplementary material for the perceived age classifier (top 4 attributes). The verbal description study combines a mixture of attributes from our and the authors' models, explaining Face and Cats/Dogs classifiers. Results for both studies were provided by 30 users (different per study).

Table 1 shows that the results we obtain are within a standard deviation of their results; verifying their contribution that StylEx provides attributes that are easily distinguishable by humans.

Table 3 depicts the three most common words used, to describe the most prominent attribute that changes in the images (see Appendix D). By inspecting the results, we draw two main conclusions. First, for all coordinates except skin color (i.e. 5[th] row in Face(age/gender) classifiers), the majority of the users use the same word in their descriptions. Second, the most common word used is different per attribute, proving that each attribute is unique. Our results agree with the results provided in the original paper.

## 6.4 Reconstruction Generalization

To further investigate the proposed model, we create new latent variables using images from the FFHQ dataset on our architectures with their pre-trained weights. Then, we use the obtained latent variables to reconstruct the images using our pre-trained generator. Finally, we follow the same process using their architecture and compare the resulting images. Our StylEx reconstructs a clearer image, compared to their model which is more blurred. This may occur because of some differences in the formatting between the frameworks.

|                   | Theirs          | Ours            |
|-------------------|-----------------|-----------------|
| Perceived Gender  | $0.96(\pm0.047)$ | $0.94(\pm0.031)$ |
| Perceived Age     | $0.983(\pm0.037)$ | $0.978(\pm0.025)$ |

Table 1: **Classification study results.** Correct identification of the top-6 attributes.

## 6.5 Training

The training proved quite volatile. The $\mathcal{L}_{rec}$ would get stuck in local minima during training. Examples of the images reconstructed by the fully trained model (see Appendix E).

Lang et al. experimented with two training regimens. The first regimen was trained using only $E(x)$ as $w$, the inputs to the generator, and the above loss. The second regimen alternated between using $E(x)$ and a randomly generated encoding, $\bar{w}$. This $\bar{w}$ is created by applying a mapping network to $z$, where $z \sim \mathcal{N}(0^n, 1^n)$ and $n$ is the dimensionality of $w$. For this randomly generated $\bar{x}' = G(\bar{w})$, only the adversarial loss is calculated. Training using $\bar{w}$ can be viewed as the same as training a vanilla StyleGAN. Because we are unsure which method was used for the results in their paper and notebook, we experimented with both. However, the first regimen was the only one that converged.

Though we were able to train a model, due to time constraints, we were unable to fully investigate Claim 1.

Again due to time constraints, we were unable to run AttFind on the trained model to fully test Claim 3.

# 7 Discussion

Using the definition of reproducibility[3] by the U.S. National Science Foundation (NSF) subcommittee on replicability in science, it is difficult to determine Lang et al.'s reproducibility. All details regarding the experimental setup, such as the hyperparameters, the hours of training, the number of steps, the labels of the datasets, etc. are omitted, thus recreating the exact materials of the original investigators is difficult. Since our definition is an implication and we cannot satisfy the first condition, we cannot determine the reproducibility.

Instead, we will use a looser definition of reproducibility. We will refer to reproducibility as the ability for another researcher to test their claims. We found that, given enough time, the StylEx is seemingly reproducible. However, given a limited time budget such as our own, the paper is not fully reproducible. We, therefore, can only provide unit tests of their claims. The following sections will discuss information from the results section 6 and to what degree they confirm reproducibility claim by claim.

## 7.1 Claim 1

The most difficult claim to investigate, given a limited time budget, is the effect of classifier-based training on the StyleSpace. The original paper trains three models, the StylEx with and without integration of the classifier in training and the StyleGAN v2. We found, once the training algorithm is implemented correctly, just training all three models will take at least 24 hours for 50,000 epochs on one GPU even for the simple MNIST dataset. The authors stated that it took approximately a week to train StylEx with 8GPUs. Over two weeks of training time is beyond our time constraints.

In addition, we observed that training is volatile.[4] The reconstruction error stagnates in a local minimum before suddenly dipping. However, the model was not always able to escape the local minima within 50,000 iterations. This suggests that, though their results are likely replicable, their replicability may be stochastic. This again hinders reproducibility when time is limited.

## 7.2 Claim 2

The claim that the authors document the most was Claim 2, their AttFind method. Because the method was implemented in the notebook provided, testing reproducibility was easy.

---

[3]"reproducibility refers to the ability of a researcher to duplicate the results of a prior study using the same materials as were used by the original investigator"

[4]An example of successful training can be found here and one where the model failed to converge here

We were able to verify that for the perceived age classifier, our model obtains the same top attributes. We conclude that their method can discover the most influential classifier-related attributes.

In addition to their notebook, we modified the AttFind method to find the principal attributes of a single image as shown in Figure 2. This validated the sub-claim of AttFind that StylEx can provide image-specific explanations. Rather than finding the *globally* important attributes, the model can find the *locally* important attributes for a particular image.

### 7.3 Claim 3

The authors claim that StylEx is *applicable* to a variety of real-world problems. Applicability can be interpreted in two different ways. One can interpret it as being *possible* to apply StylEx to a variety of domains, or as *practical* to apply StylEx to a variety of domains. From what we have seen in Figures 1, 2, it is possible to use StylEx for explaining an age classifier, thus it can explain a real-world problem. From Figure 6 (see Appendix E), we found that the StylEx can be trained to, at minimum, reconstruct MNIST data, thus multiple domains.

Though we have found that it is *possible*, we have also found that it is seemingly impractical. Every domain requires the model to be retrained, meaning every domain requires days or weeks of training.

### 7.4 What was easy

The open-source notebook is very well structured, which combined with the pseudo-code outlined in Algorithm 1 of their paper, made the AttFind method easy to replicate. In addition, the provided pre-trained models helped to derive some of the vague components of StylEx model.

### 7.5 What was difficult

As we already emphasized, there are many difficulties in reproducing this paper. StylEx is built on top of several previous papers making the knowledge needed for implementation substantial. Lang et al. proposed a model without providing code, that is computationally expensive, and with volatile training behavior. In addition, that is sensitive to hyperparameters, which in our case were unknown. Even when scaling down the complexity of the model using smaller resolutions, the time cost of training exceeds what was feasible with our time constraints.

Taking shortcuts to subvert these difficulties had a multitude of challenges. We found loading weights from TensorFlow to PyTorch deceptively complex and far from trivial due to differences between the frameworks. Even evaluating their notebook came with difficulties as the dataset they trained on FFHQ does not officially have labels, so the details of their dataset were unknown.

### 7.6 Future Work

The primary goal of this paper was to reproduce the work of Lang et al., however, through reimplementing their code, we found two open avenues for future research. Firstly, the paper focused on general image explanations but did not show examples of misclassified data. It would be interesting to see what insights can be obtained through StylEx. Secondly, the paper compared StylEx only with StyleGAN v2 models. AttFind seems applicable to general autoencoders, and not specific to GANs. Viewing StylEx as an autoencoder, rather than a GAN seems like a promising angle for scalability to a similar counterfactual generator.

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

## A  Our StylEx vs Lang et al.'s

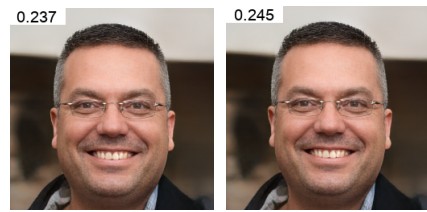

(a) TensorFlow (theirs)  (b) PyTorch (ours)

Figure 3: **Comparison of StylEx models results.** The probabilities shown correspond to being classifier as young.

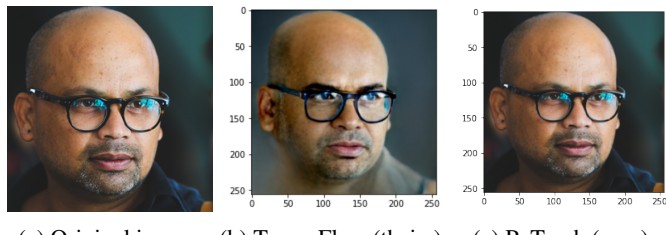

(a) Original image  (b) TensorFlow (theirs)  (c) PyTorch (ours)

Figure 4: **Comparison of StylEx models encoding and then reconstructing an image.** Both models use their encoder and classifier to produce the latent variable. Then using their generator the image is reconstructed from the latent variable.

## B  AttFind Lang et al.'s top attributes

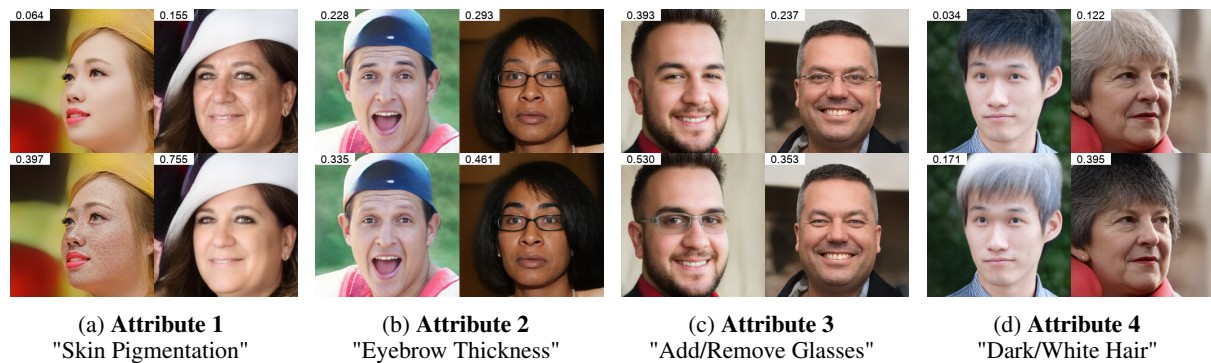

(a) **Attribute 1**
"Skin Pigmentation"

(b) **Attribute 2**
"Eyebrow Thickness"

(c) **Attribute 3**
"Add/Remove Glasses"

(d) **Attribute 4**
"Dark/White Hair"

Figure 5: **Top 4 attributes for the perceived age classifier detected by Lang et al.'s pre-trained model.** These images show how the probability of classifying a person as young or old changes based on each attribute. On the first column of each image, we display the probability of the person being classified as old and on the second column the probability of them being classified as young.

# C Hyperparameters

| | Our StylEx | Lang et al's StylEx |
|---|---|---|
| Step Size | 1e-3 | 2e-4 |
| Number of Steps | 50,000 | 250,000 |
| Total Loss Weights ($\mathcal{L}_{rec}, \mathcal{L}_{adv}, \mathcal{L}_c, \mathcal{L}_{PL}$) | 1,1,1,1 | 1,1,1,? |
| Reconstruction Loss Weights ($\mathcal{L}_w, \mathcal{L}_x, \mathcal{L}_{LPIPS}$) | .1, 1, .1 | .1, 1, .1 |
| Latent Dimension | 32 | 512 |
| Number of Classes | 2 | 2 (depending on data) |
| Image Resolution | 32 | 256 |
| Classifier Structure | DenseNet121 | MobileNet |
| Optimizer | Adam | ? |

Table 2: Training hyperparameters

# D Verbal Description Study

| Cats/Dogs | | | |
|---|---|---|---|
| | eye: 0.73 | pupil: 0.16 | shape: 0.1 |
| | mouth: 0.73 | open: 0.3 | tongue: 0.16 |
| | ear: 0.90 | right: 0.06 | become: 0.06 |

(a)

| Face | | | |
|---|---|---|---|
| | eyebrow: 0.90 | thick: 0.17 | brow: 0.07 |
| | tooth: 0.30 | lip: 0.10 | disappear: 0.07 |
| | glass: 0.90 | size: 0.13 | bigger: 0.10 |
| | mouth: 0.70 | open: 0.40 | lip: 0.10 |
| | bright: 0.37 | skin: 0.30 | light: 0.27 |
| | mustache: 0.93 | facial: 0.07 | hair: 0.07 |
| | eye: 0.77 | color: 0.47 | eyelash: 0.13 |

(b)

Table 3: **Verbal description study results.** The 3 most common words used in user descriptions for the Cat/Dogs (a) and Face (age/gender) (b) classifiers. This user study proves the distinctness of each attribute since the most common word used to describe each attribute change is different per classifier.

 # E   MNIST Reconstruction

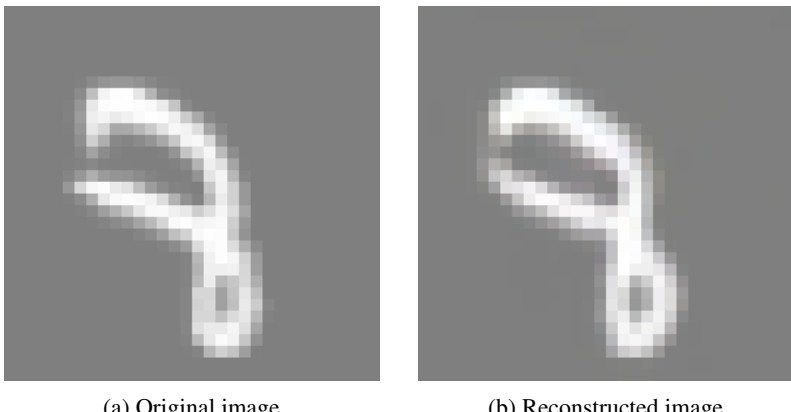

(a) Original image          (b) Reconstructed image

Figure 6: An example of image reconstruction on the MNIST dataset. The StylEx had converged however, it was trained conditioned on a classifier that always predicted 8, thus was effectively trained without a classifier. It's loss curves can be found here.

