# OpenReview forum: "Reproducibility study for "Explaining in Style: Training a GAN to explain a classifier in StyleSpace""
_ML_Reproducibility_Challenge/2021/Fall — RC2021_

### Official Review · Reviewer_x9hs · 2022-02-27
**Solid reproducibility effort that adds valuable insights about the original paper**

**Rating:** 7
**Confidence:** 4

**Review:**

**Reproducibility summary:** The summary is clear and well summarizes the authors' reproducibility effort.

**Scope of reproducibility:** The authors verify the following three claims made by the original paper: 1) The StylEx model can capture classifier-specific attributes. 2) The AttFind algorithm can find important style space coordinates for the classifier, which can then be used to generate counterfactual explanations. 3) The StylEx model is applicable for explaining a large variety of classifiers and real-world domains, and can provide explanations understood by human users.

**Code:** The authors reimplement the AttFind algorithm in PyTorch and reverse engineer the StylEx model. They use the original authors’ pre-trained model weights to test their reconstruction.

**Communication with the original authors**: The authors have corresponded with the original authors and received additional details about the encoder architecture and some of the experimental setup.

**Hyperparameter search:** The authors do not conduct hyperparameter search and give two reasons for this. However, they provide a complete list of hyperparameters they used which can help future users of this work.

**Ablation study**: The authors did not conduct any ablation studies.

**Discussion on results:** The authors state that it is difficult to determine the original work’s reproducibility according to NSF subcommittee on replicability in science’s standards. However, using a looser definition of reproducibility (ability for another to test the original work’s claims), the authors state that StylEx is seemingly reproducibile given enough time and resources. They provide a detailed discussion of how their investigation supports (or doesn’t support) the original paper’s claims.

**Recommendations for reproducibility:** The authors don't provide explicit recommendations. However, their descriptions of the difficulties they ran into may help the original authors improve the reproducibility of the work and/or future researchers building on this work

**Results beyond the paper**: There are no results beyond the original paper.

**Overall organization and clarity:** Overall, the report was organized and clearly written.

---

### Official Review · Reviewer_sTG6 · 2022-03-01
**Well-written and well-structured replication study with insightful findings**

**Rating:** 9
**Confidence:** 4

**Review:**

The paper presents results for replicating StyleEx. The paper is well structured and pleasant to read, as it provides sufficient introduction and background to the original method, and thoroughly describes how the original code was used and extended to replicate the results. The claims are concise and sensible.

The authors did reuse the code provided from the original paper, but had to extend it due to missing pretrained models (e.g. by reverse engineering the model architectures).

The proposed paper did not evaluate on all originally used datasets due to time limitations, which is understandable. The reported results are still of value. The paper was able to confirm the stipulated claim 2 wrt finding classifier-related attributes in the StyleSpace. The residual claims were only partially confirmed, giving an interesting perspective on the available constraints due lack of provided resources. Lastly, the authors give valuable insight into potential problems, such as the volatile training process, as well as missing details in the original paper on the conducted experimental setup.

---

### Meta-Review · Program_Chairs · 2022-04-07

**Recommendation:** Accept
**Confidence:** 4

**Metareview:**

The authors went above and beyond the original study by extending the code due to missing pretrained models and confirming the different claims of the original paper and providing useful insights into them. This is a worthy and interesting contribution to the reproducibility challenge!

---

### Decision · Program_Chairs · 2022-04-09

**Decision:**

Accept

**Comment:**

Following the recommendation of reviewers and meta-reviewer, the paper is accepted for ML Reproducibility Challenge 2021, and will be published in the upcoming special edition of ReScience Journal.